# Processing GPR Surveys in Civil Engineering to Locate Buried Structures in Highly Conductive Subsoils

Rosendo Mendoza [1], Carlos Araque-Perez [2], Bruna Marinho [3], Javier Rey [3,*] and Mari Carmen Hidalgo [3]

1   Mechanical and Mining Engineering Department and CEACTEMA, Higher Polytechnic School of Linares, Technological Scientific Campus, University of Jaen, 23700 Linares, Spain; rmendoza@ujaen.es
2   Andalusian Institute of Geophysics, Campus de Cartuja, University of Granada, 18071 Granada, Spain; carlos.araque@ugr.es
3   Geology Department and CEACTEMA, Higher Polytechnic School of Linares, Technological Scientific Campus, University of Jaen, 23700 Linares, Spain; bmb00013@red.ujaen.es (B.M.); chidalgo@ujaen.es (M.C.H.)
*   Correspondence: jrey@ujaen.es

**Abstract:** Many studies have illustrated the great benefit of ground-penetrating radar (GPR) in civil engineering. However, in some cases, this geophysical survey method does not produce the desired results due to the electromagnetic characteristics of the subsoil. This study presents the results obtained in two locations near Linares (southern Spain), evaluating the detection of structures buried in conductive host materials (0.02 S/m in site 1 and 0.015 S/m in site 2) characterized by strong signal attenuation. Accounting for the study depth, which was 1.5 m, a 500 MHz shielded GPR antenna was used at both sites. At the first site, a controlled experiment was planned, and it consisted of burying three linear elements. An iron pipe, a PVC pipe, and a series of precast blocks were buried at a depth of 0.5 m in a subsoil composed of highly conductive clayey facies. To eliminate additional multiples caused by other superficial structures and increasing the high-frequency content, the predictive deconvolution flow was applied. In the 3D processing, the cover surfaces technique was used. Once the acquired GPR signals was analyzed and the optimal processing flow established, a second site in which different infrastructures in a conductive host medium formed by marly facies was explored. The 2D flow and 3D processing applied in this work allows to detect and see the continuity of some structures not visible for the default processing.

**Keywords:** non-destructive survey; ground-penetrating radar; civil-engineering applications; GPR signal attenuation

## 1. Introduction

### 1.1. Theoretical Overview

Ground-penetrating radar (GPR) is a geophysical prospecting technique that is based on the study of the propagation of electromagnetic waves (EM) through the subsoil. This wave aspect means that it is a high-resolution method that operates at "high frequencies (>10 MHz) or in a low-loss state when the product between the conductivity ($\sigma$) and the dielectric constant ($\varepsilon$) is less than 1 (loss factor P << 1, known as loss tangent)" [1]; thus, the velocity of the electromagnetic waves does not depend on the frequency but only on the electromagnetic response of the medium in which they travel [1–33]. The GPR signals are affected by a series of characteristics intrinsic to the materials, such as the composition, texture, degree of saturation, and composition of the interstitial fluids [2–5]. These two aspects (resolution and response to small variations in the environment) make GPR exploration a widely used method in surface geoscience studies, including studies focused on surface geology, hydrology, mining, environment, forensics, and archaeological or heritage studies [5–15].

In civil engineering, this technique has been successfully applied to evaluate the state of pavements, roads, bridges, tunnels, the location of utilities buried in the substrate

and hidden structures in buildings as well as in the characterization and diagnosis of construction materials or the detection of humidity [16–23].

An algorithm that automatically detects and segments small cracks in asphalt pavement at the pixel level has been described [24]. The research results represent a further step toward accurately detecting and characterizing internal vertical cracks in asphalt pavement. This application is practical for considering subsequent maintenance work due to the use of the pavement [25]. A novel YOLOv3 model identifying and localizing concealed cracks in asphalt pavements through GPR was proposed [26,27]. An overview from 2015 to the present of the states of research on employing the GPR technique in the civil engineering was provided [28], in which promising directions for future research were identified. More directly related to the line of this research, there are also numerous studies that have applied GPR for the detection of buried services, basically cables and pipes. The technique allows locating the position and depth of these services as well as locating possible leaks or illegal taking of supplies [17,29–31].

However, despite the many examples that demonstrate its effectiveness, there are situations, generally unpublished, where this method fails to produce the expected outcomes. Fundamentally, these situations are due to two causes: (i) when there is little contrast between the structure to be detected and the host medium or (ii) when the host material is conductive and/or has high magnetic susceptibility [2,5,22,32].

In an undulatory wave motion and in the first approximation [2,5,33–35], the propagation equation of the electric field ($E$) for a plane wave is obtained as follows:

$$\vec{E} = f(\beta \pm vt)e^{\mp\alpha\beta} \tag{1}$$

where $f$ is a multivariable function in which $\beta$ represents the distance in the direction of propagation, $v$ represents the velocity of propagation, and $\alpha$ represents the attenuation coefficient as the parameter describing the decay of the wave energy with the propagation distance through a host media. For low-loss and diamagnetic materials [33–39], when P << 1 and considering that the loss factor is equal to the ratio between the conductivity ($\sigma$) and the product of the angular frequency ($\omega$) with the permittivity ($\varepsilon$), P = $\sigma/\omega\varepsilon$, the above parameters are expressed as follows:

$$\beta = \vec{r} \cdot \hat{\kappa} \tag{2}$$

$$v = \frac{c}{\sqrt{\varepsilon_r\mu_r}}\left(\frac{\sqrt{1+P^2}+1}{2}\right)^{-\frac{1}{2}}, \text{ if } \sigma << \omega\varepsilon \text{ and } \mu_r \approx 1, \text{ then resulting in } v = \frac{c}{\sqrt{\varepsilon_r}} \tag{3}$$

$$\alpha = \omega\frac{\sqrt{\varepsilon_r\mu_r}}{c}\left(\frac{\sqrt{1+P^2}-1}{2}\right)^{\frac{1}{2}}, \text{ if } \sigma << \omega\varepsilon \text{ and } \mu_r \approx 1, \text{ resulting in } \alpha = \frac{1}{2}\sigma\sqrt{\frac{\mu}{\varepsilon}} \tag{4}$$

where $\vec{r}$ is the position vector, $\hat{\kappa}$ is the direction and magnitude of EM wave propagation, $\varepsilon_r = \frac{\varepsilon}{\varepsilon_0}$ is the relative dielectric constant or relative permittivity, $c$ is the speed of light in a vacuum, and $\sigma$ and $\mu$ represent the conductivity and magnetic susceptibility (also known as permeability) of the material, respectively; $\mu_r$ is the relative permeability, and $\alpha$ the attenuation coefficient.

Equation (3) describes the dependence of the EM wave propagation velocity on the relative dielectric constants of the materials, establishing an inverse dependence. This relationship implies variations of up to an order of magnitude, where the extreme values are located between the air ($\varepsilon_r$ = 1) and water ($\varepsilon_r$ = 81). Equation (4) describes the attenuation coefficient as the decay of the EM's energy as a function of the constitutive parameters. However, in the attenuation of an EM signal, other attenuation phenomena, such as diffraction, scattering, and chromatic dispersion, should be incorporated [5,33–35]. In this framework, the penetration of the EM wave is parameterized by the skin depth ($\delta$), which

is an important concept in EM theory and is used in GPR to describe the characteristic distance at which the amplitude of an electromagnetic wave significantly decreases as it propagates through a conducting or semiconducting host media [33–35], as it is an energy attenuation factor of the EM study due to its inverse relationship with the attenuation coefficient [1,3,33]. Normally, the skin depth refers to the distance at which the amplitudes are reduced by a factor of 1/e, that is, 36.8% of the original signal, and can be obtained as follows:

$$\delta = \frac{1}{\alpha} \text{ resulting in}: \ \delta = \begin{cases} 503\sqrt{\dfrac{1}{\sigma f}} & \text{for } \omega\varepsilon << \sigma \ (\text{first condition}) \\ 0.0053\sqrt{\dfrac{\varepsilon_r}{\sigma}} & \text{for } \sigma << \omega\varepsilon \ (\text{second condition}) \end{cases} \tag{5}$$

From the above equation, there are two conditions for the skin depth. The first one is when the product between the irradiated frequency and the dielectric constant is much smaller than the conductivity of the host media. In this case, the skin depth depends on the frequency of the device used, as is the case in studies performed with inductive methods, such as frequency domain EM (FDEM) [1,3]. In contrast, for the second condition, the conductivity is less than the product between the irradiated frequency and the dielectric constant giving rise to an equation that is calculated independently of the frequency of the device used. This is the case for non-inductive methods, such as ground-penetrating radar [1,3]. Therefore, for conductive media the skin depth tends to be low, increasing the attenuation coefficient of the EM signal.

The objective of this study is to evaluate signal detection in host media with a low skin depth, leading to materials such as conductive media near the electromagnetic wave state boundary for GPR surveys.

### 1.2. Study Areas

Studies were carried out at two sites near Linares (southern Spain, Figure 1A). The first zone (site 1) was selected in the vicinity of the city's university campus (Figure 1B), which has a very clayey substrate. Three linear elements were buried (a PVC pipe, an iron pipe, and precast blocks), all of them at a depth of 0.5 m. In this case, knowing the buried structure, its position, and its depth is intended to verify the ability of the technique to detect them. The second one (site 2) consists of marly subsoil where urbanization works were carried out years ago (Figure 1C), but they were abandoned before their completion. In this second case, the goal is to determine the technique's ability to detect the buried utilities associated with this urbanization, as there is no information on the existing structures under the substrate.

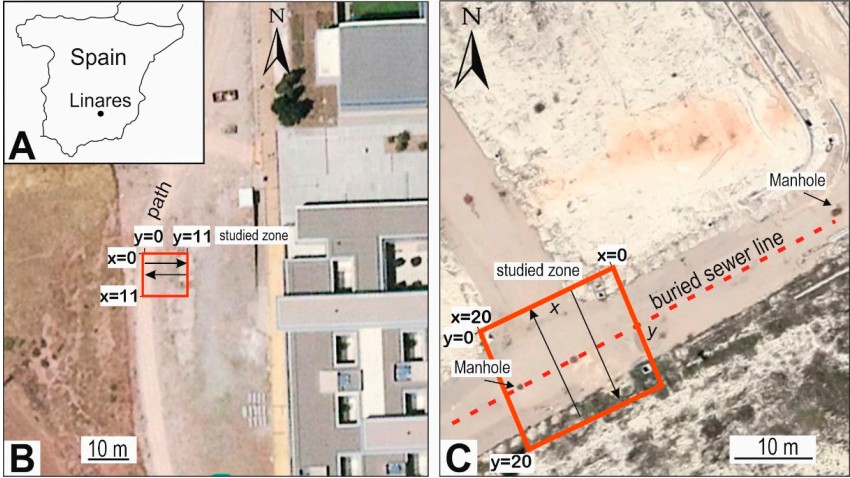

**Figure 1.** (**A**) Geographic location of the studied areas. (**B**) Aerial photograph showing the position of site 1, where the red square represents the controlled test, at the university campus. (**C**) Aerial photograph showing the rectangle placement of site 2 with industrial infrastructures.

## 2. Materials and Methods

### 2.1. Experimental Design and Data Acquisition

At the first field experimental site (university campus; Figure 1B), three parallel trenches approximately 0.5 m wide, 4 m long, and 0.7 m deep were constructed (Figure 2A). The top of these three buried elements (Figure 2A–C, respectively) should be detected at approximately 0.5 m depth, and the average thickness of the structures is approximately 0.2 m. A PVC pipe was placed in the first trench to the east, next to one of the pre-existent concrete slabs (Figure 2A). In the second trench, the precast blocks were installed (Figure 2B). Note that these bricks have voids inside them. Finally, in the third trench, the iron pipes were placed (Figure 2C).

In site 1 (university zone), a square of 11 m × 11 m was planned for detecting the three buried structures described above. They also incorporated part of two old, highly degraded concrete slabs. Figure 3A shows the spread GPR chart with the location of buried elements and the cemented portions.

In site 2 (old, abandoned urbanization; Figures 1C and 2D), some utilities were buried, such as the infrastructure for water, gas, sewer, electrical, and telephone facilities, but this information is not available (Figure 2D).

The two surveys were carried out in September 2022 on very dry lithologies. In both, a shielded antenna of 500 MHz (e RAMAC/GPR system, Pro-Ex model, by MALA GEOSCIENCE) was used (Figure 2D). This shielded antenna is a bow-tie design with an approximate maximum penetration depth of 6 meters. It provides a radial resolution of about 0.04 meters in the main radiation lobe for an environment with typical soil conditions where high-conductivity layers are absent. To define the parameters used in this work, several preliminary GPR campaigns were carried out, varying the time window (TW), trace spacing, and profile spacing. This previous work made it possible to define the parameters that offered the best response. Table 1 summarizes the main acquisition parameter in the two sites, whose acquisition geometry is shown in Figure 3.

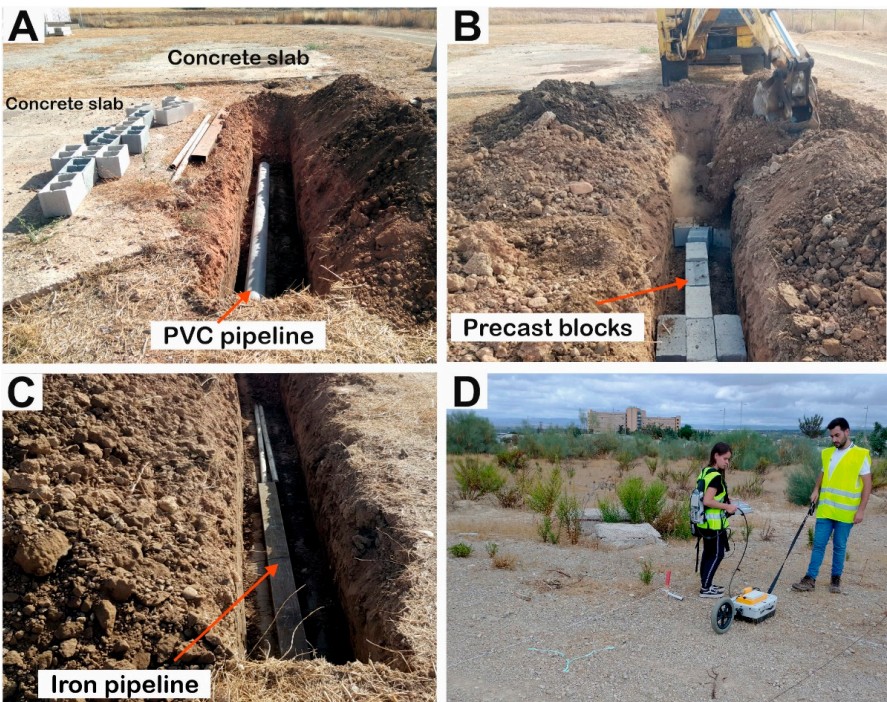

**Figure 2.** (**A**) Views of the buried elements used in the tests carried out in site 1. PVC pipe was placed in the first trench to the east. (**B**) Second trench and the precast blocks installed. (**C**) Third trench, where the iron pipe was installed. (**D**) General view of the site 2: remains of old manholes and the direction of two perpendicular streets are observed. All data were acquired with a 500 MHz shielded antenna.

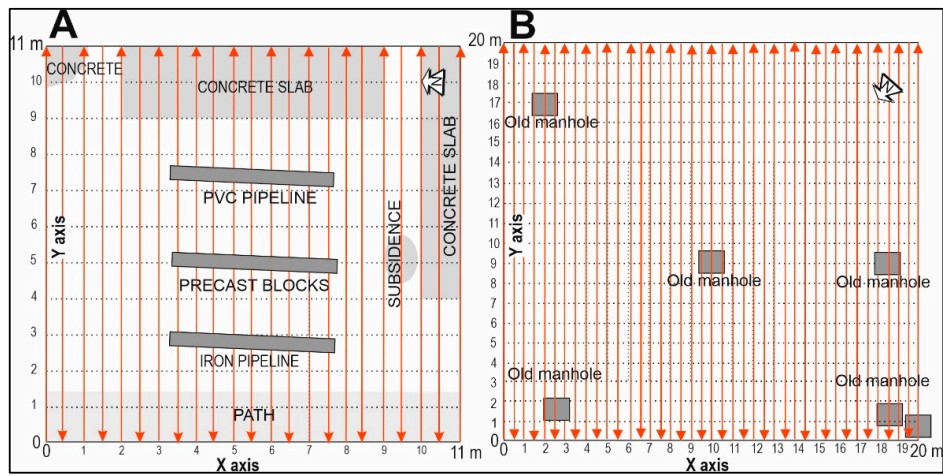

**Figure 3.** (**A**) Layout of the acquisition geometry in site 1 with the location of the controlled buried elements and the coverage of acquisition GPR profiles. (**B**) Layout of the acquisition geometry in site 2 with the location of the superficial elements and the GPR profiles coverage.

**Table 1.** Main GPR acquisition parameters.

| Main GPR Acquisition Parameters | | |
|---|---|---|
| **Parameters** | **Site 1** | **Site 2** |
| Profile spacing | 0.5 m | 0.5 m |
| Number of profiles | 23 | 41 |
| Trace spacing | 0.01 m | 0.01 m |
| Time window | 30 ns | 40 ns |
| Samples trace | 248 | 312 |

*2.2. Equipment*

The experiment was performed with a 500 MHz shielded antenna from MALA Geosciences. This antenna consists of a shielded bow-tie design modified by the manufactured to avoid signal noise from external sources, such as mobile telephones, FM broadcast, television, and radar system. The shielded antenna means that most of the energy is only transmitted in one direction. The shielded antenna comprises both transmitter and receiver antenna elements in one single housing. The antenna is calibrated by the manufacturer to generate emissions below the limits established by Federal Communications Commission (FCC15.509 and FCC15.209) [40,41] and American National Standard Institute (ANSI C63.4) [42]. In this instance, the manufacturer had previously conducted multiple tests under antennas with the same characteristics. Based on these tests, the manufacturer provided information on the radiation patterns [43,44]. This includes a 500 MHz-centered main emission lobe, a pulse emission of 10.40 ns, a 3dB bandwidth of 361 MHz with an approximate maximum penetration depth of 6 m, and a radial resolution of 5 cm in typical soil conditions where high-conductivity layers are absent. Figure 4A,B show an idealized radiation pattern of the 500 MHz antenna. The main lobe is centered at 500 MHz, and the side lobes are around 200 MHz and 700 MHz.

With the above information, the theoretical signal attenuation of the antenna in clay media at 0.5 m could be calculated through the attenuation coefficient ($\alpha$), considering the 500 MHz central frequency (Equation (6)) [45–47], a dielectric constant ($\varepsilon$, permittivity) for clay media of ~$7.0 \times 10^{-11}$ F/m (approximated to the permittivity of free space), and a magnetic susceptibility ($\mu$, permeability) of ~1.0 H/m (approximated to the permeability of free space). The skin depth ($\delta$) for clay is generally very low and can be considered negligible when calculated, such as tan ($\delta$). Finally, the theoretical value of the attenuation

coefficient ($\alpha$) at the average frequency at 0.5 m is 0.026 Np/m, which is approximately equal to 0.226 dB/m.

$$\alpha = \omega \sqrt{\mu\varepsilon \tan\delta} \tag{6}$$

Therefore, despite the fact that Equation (6) includes frequency as a parameter for calculating the attenuation coefficient, it is more appropriate to use Equations (4) and (5), which are approximations made within the EM theory for the frequency range from 40 MHz to 5 GHz, which refer to light speeds and low-loss state. As can be inferred from these equations, regardless of the fact that the radiated frequency is a crucial factor in depth penetration and resolution, the signal losses (like the attenuation coefficient) do not depend on the frequency but rather on the properties of the host media [1–5,33].

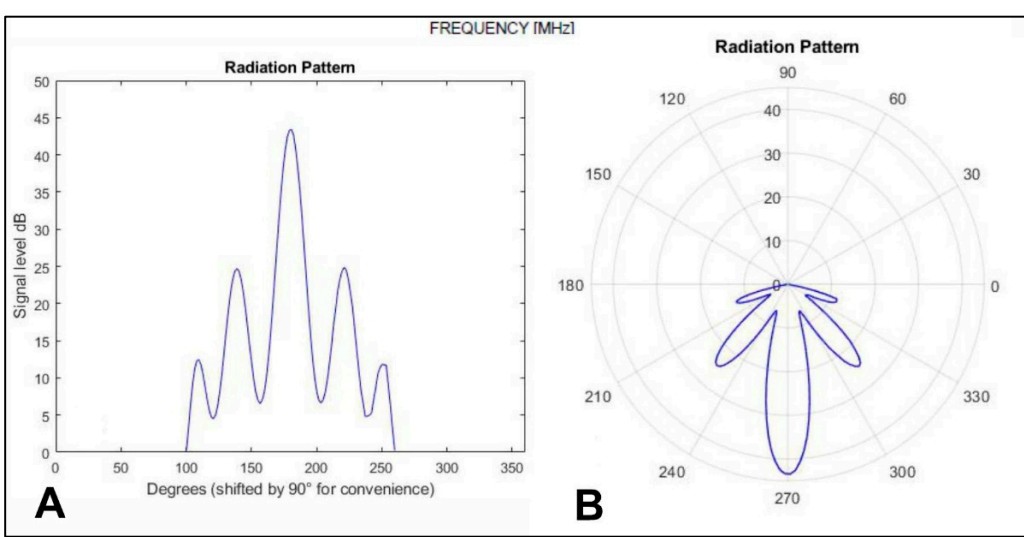

**Figure 4.** (**A**,**B**) Idealized radiation pattern of the 500 MHz antenna constructed from the manufacturer emission test. The main lobe is centered at 500 MHz, and the side lobes are around 200 MHz and 700 MHz.

It is important to notice that attenuation coefficient and losses are related concepts in GPR systems, but they represent different aspects of signal propagation and signal strength reduction in the subsurface medium. On the one hand, the attenuation coefficient ($\alpha$) in GPR refers to the rate at which the electromagnetic wave's energy decreases as it propagates through the subsurface medium. It is a measure of how much the signal weakens over a unit distance of propagation. The attenuation coefficient is usually expressed in units of Nepers per meter (Np/m) or decibels per meter (dB/m). The attenuation coefficient accounts for factors such as absorption, scattering, and reflection of the electromagnetic waves as they interact with the subsurface materials. Different materials have varying attenuation coefficients, and they can change with frequency and moisture content. For instance, dry clay soils have a theoretical $\alpha$-value that spans from 0.06 to 6 dB/m, whereas wet clay produces values between 6 and 50 dB/m. Higher attenuation coefficients indicate stronger signal absorption and scattering, leading to faster signal weakening as the wave travels through the medium [3,33,48–51].

On the other hand, losses in GPR systems refer to the other mechanisms that cause a reduction of signal strength during its transmission, reception, and processing. These losses are caused by various factors, e.g., geometric expansion of wavefront, transmission and reflection losses, scattering losses, absorption losses, and antenna losses, among others [3,7,33,48–51].

Therefore, the attenuation coefficient quantifies the rate of signal weakening due to the subsurface medium's properties, while losses encompass a broader set of factors that result in signal reduction during its propagation and measurement in the GPR system.

### 2.3. GPR Data Analysis of Site 1 to Stablish the 2D Processing Flow

Raw GPR data were processed using two codes: the Reflexw software, version 7.0 [38] and GPR-SLICE (https://www.gpr-survey.com/) that is accessed on 25 January 2023 [34] (see Table 2 and Figure 5). As a first starting point, in order to establish the processing flow, the effective skin depth (mean) in site 1 was performed utilizing Equation (5) for clay and marl media. These lithologies are characterized by a dielectric constant spanning from 10 to 40 and a host medium velocity ranging between 4.7 and 9.5 cm/ns, reaching a skin depth ranging from 1.0 to 1.67 meters, with attenuation coefficient between 5 and 9 dB/m when employing Equation (4). The values were originally provided in Nepers per meter and subsequently converted into decibels per meter (Figure 6D). In this instance, the velocity of the host media was calculated based on the known fact that there are three buried objects at 50 cm depth, which are detected at around 20 ns in the B-scans. Using the classic equation $v = \frac{x}{t}$, a velocity of 5 cm/ns was calculated, which was then corrected by knowing that the air space between the antenna and soil travels at 4 ns. Hence, the effective time within the substrate is 16 ns, resulting in a medium velocity of 6.3 cm/ns, which corresponds to a dielectric constant of 22 (see Figure 6B). A hyperbolic analysis improved the calculation of the medium velocity and the dielectric constant, resulting in a dielectric constant of 20 and a medium velocity of 6.7 cm/ns. Based on the previously calculated values, a conductivity value of 0.02 S/m was obtained by interpolating between the limits of the dry and wet clay curves depicted in Figure 6C.

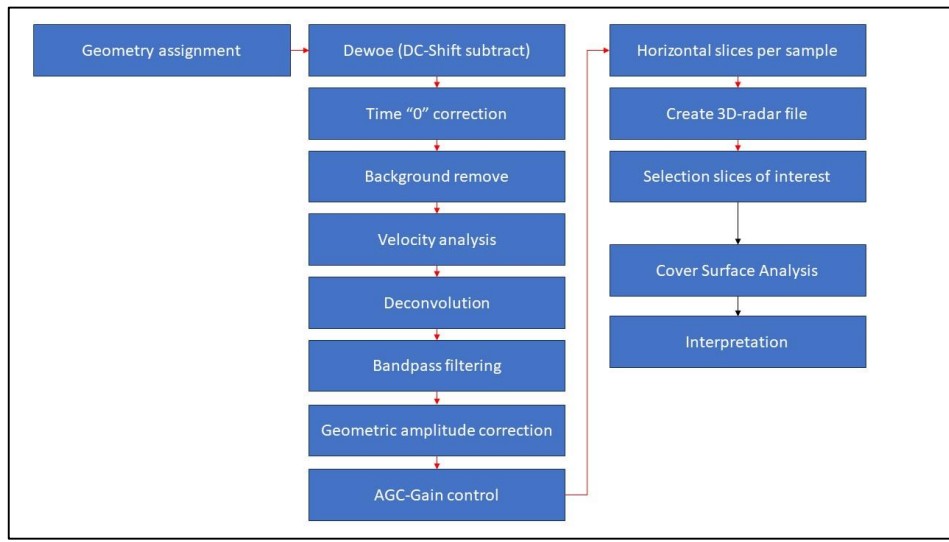

**Figure 5.** General GPR processing work-flow chart used in site 1 and site 2.

**Table 2.** Planned 2D processing flow for site 1—controlled experiment.

| Steps | Description |
|---|---|
| Subtract DC shift | To eliminate the residual voltage |
| Static correction | To adjust the delay time at time = 0 ns |
| Remove the background noise | To eliminate the multiples proceed by coupled wave (air-surface) |
| Spiking predictive deconvolution | To eliminate additional multiples caused by other superficial structures and increase the high-frequency content; Operator length of 21 ns, predication lag of 7 ns, pre-whitening 10% |
| Bandpass filter | 50 MHz–550 MHz to eliminate the high and low noise frequencies |
| Geometric amplitude correction | To restitute the amplitude content for wave front propagation |
| Gain control (GC) | To increase the amplitude between 10–30 ns where the buried structures are placed; a gain function was created by modifying the AGC |

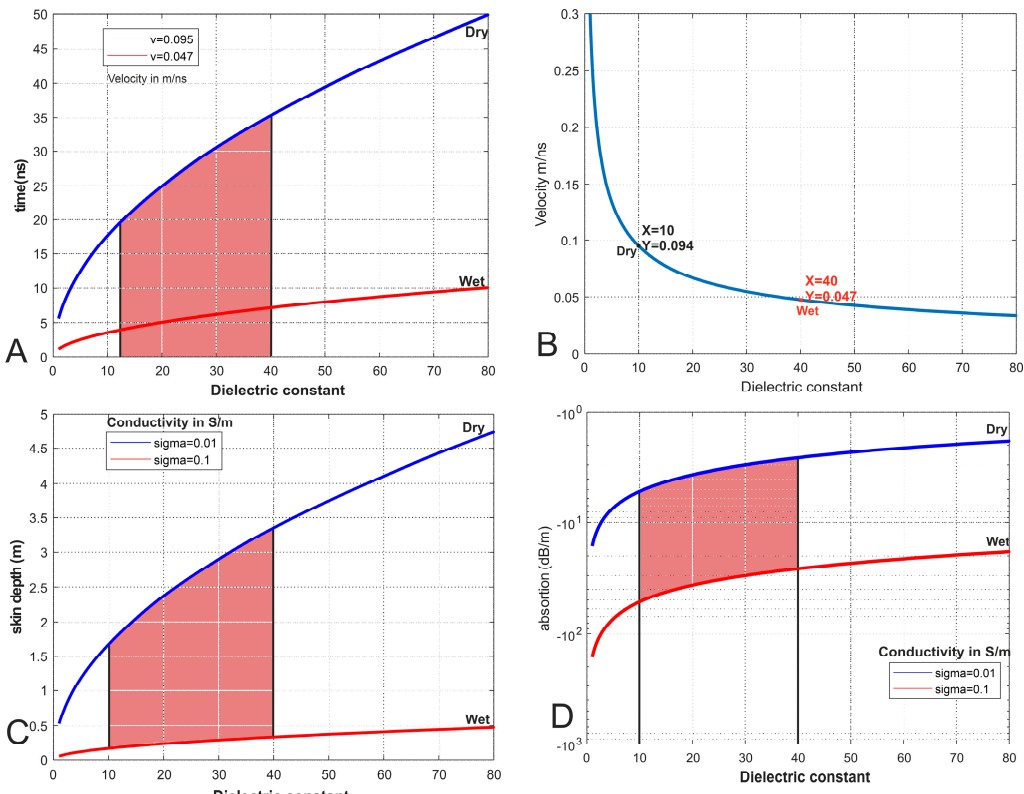

**Figure 6.** (**A**) Plot of dielectric constant and wave travel time for velocities in wet and dry clays and marls media. (**B**) Plot of velocity as a function of dielectric constant using Equation (3). (**C**) Plot of skin depth as a function of dielectric constant for conductivities in clay media using Equation (5). (**D**) Absorption plot as a function of dielectric constant for conductivities in wet and dry clay media. The calculations for all these graphic parameters consider a centrally irradiated frequency of 500 MHz. The vertical black lines indicate the limits of the dielectric constants for clays and marls in dry and wet environments, whose skin depth and absorption values at the researched sites should fall within the shaded zones.

This relationship is contingent upon the conductivity of the substratum; in this case, 0.02 S/m gives an attenuation coefficient of 6 dB/m (Figure 6D), where media with higher conductivity (moist conditions) exhibit greater dielectric constants and slower wavefront velocities, whereas drier media display the opposite trend (Figure 6A,B).

Figure 7A shows an example of a raw B-scan where the linear structures are marked with arrows; weak hyperbolas indicate their poor detection. In the same Figure, the red points are the calculated skin depth for each scan; thus, this means an effective signal up to 20 ns on average. Another advisable control test to perform on the raw data is to analyze the band of frequencies where the different GPR signals are located. In this sense, Figure 7B shows the spectral frequency power normalized for same B-scan, where the interesting GPR signals are placed between 200 MHz to 500 MHz. The maximum spectral density in the spectrum is approximately 500 MHz and was calculated for the time window of 50 ns for site 1 and 40 ns for site 2. The maximum values of the spectral density indicate the frequency with the highest concentration. During the processing flow, these spectra are used as reference to configure the bandpass filters. Notice that the amplitudes are normalized between −1 and 1 in all radargrams, and this feature has no specific physical units, such as volts or amperes. Normalization in the context of B-scans is used to adjust the signal amplitudes and emphasize amplitude variations within the image. When amplitudes are normalized between −1 and 1, a relative range of amplitudes is established based on the maximum and minimum amplitudes present in the radargram data. In this case, a value of −1 represents the minimum detected amplitude, while a value of 1 represents the

maximum detected amplitude. The normalization of amplitudes in a radargram allows for a more convenient and meaningful representation of amplitude variations in the image display. This facilitates the identification and analysis of subsurface features based on relative amplitude variations.

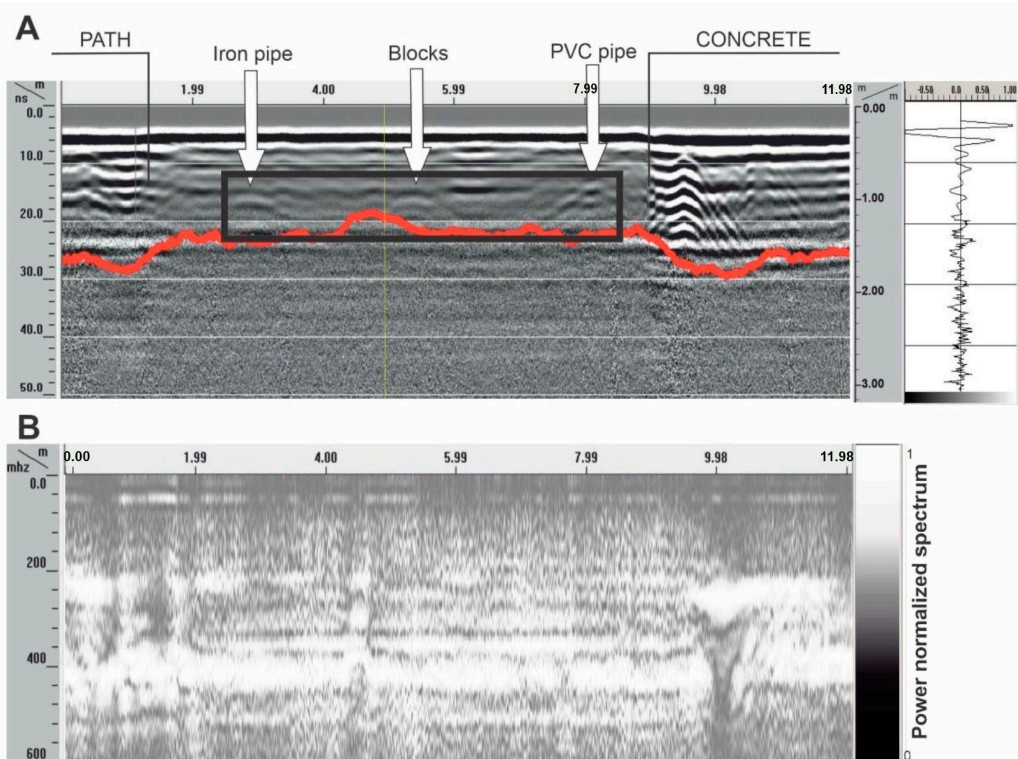

**Figure 7.** (**A**) Selected raw B-scan of the site 1 with the three buried structures (arrows). X-axis represents traces, and Y-axis represents depth in meters and nanoseconds. The image's right side depicts the A-scan (trace) indicated by the light-yellow line on the B-scan (radargram). The corresponding low signals indicate the poor detection. The red line indicates the skin-depth points for any scan; the black rectangle indicates where the structures are mainly located. (**B**) The spectral power of the same B-scan. X-axis represents traces, and Y-axis represents the frequency in megahertz. On the right site is the color scale of the power-normalized spectrum density: higher-power spectral density values are around 500 MHz (vertical scale on the left side); lower-power spectral density values are around the 300 MHz and 200 MHz bandwidth.

In conductive media with low detection, signal amplitude restitution must not only be compensated by the geometric expansion of the wavefront: additional gain corrections must also be applied due to the intrinsic absorption of the medium (Equations (4) and (5)). The above analyses indicate that the processing flow should be aimed at increasing the amplitude of the signals of interest and at "enriching" the 200 MHz–400 MHz frequency band, in which the reflections of buried elements are attributed during the experiment, as shown in Figure 7B. Furthermore, both processes must be carried out in such a way that the other reflections contained in the B-scans are affected the least. That is to say, a general increase in the amplitude of the data is not enough where the hyperbolas cannot be distinguished either, but this increase must be "selective". Another important derived piece of information is that the processing must consider that the average value of the skin depth is located almost below the three structures. Therefore, the algorithms that are applied must be very conservative in the time window of 10 ns to 30 ns in order to avoid artifacts due to the low signal-to-noise ratio (S/N) that prevails in this interval.

Considering the previous aspects, Table 2 shows the processing flow applied to raw data for site 1, while the two images in Figure 8 correspond to the effect of the processing on the previous selected B-scan. The key step of this flow was the predictive deconvolution

since with them, most of the multiples caused by other structures were eliminated, such as those located at the ends of the B-scan (path and concrete elements). In other words, the signals of interest were emphasized, minimizing the surrounding signals as much as possible. The data from site 2 (detailed in the following section) were analyzed using the same criteria.

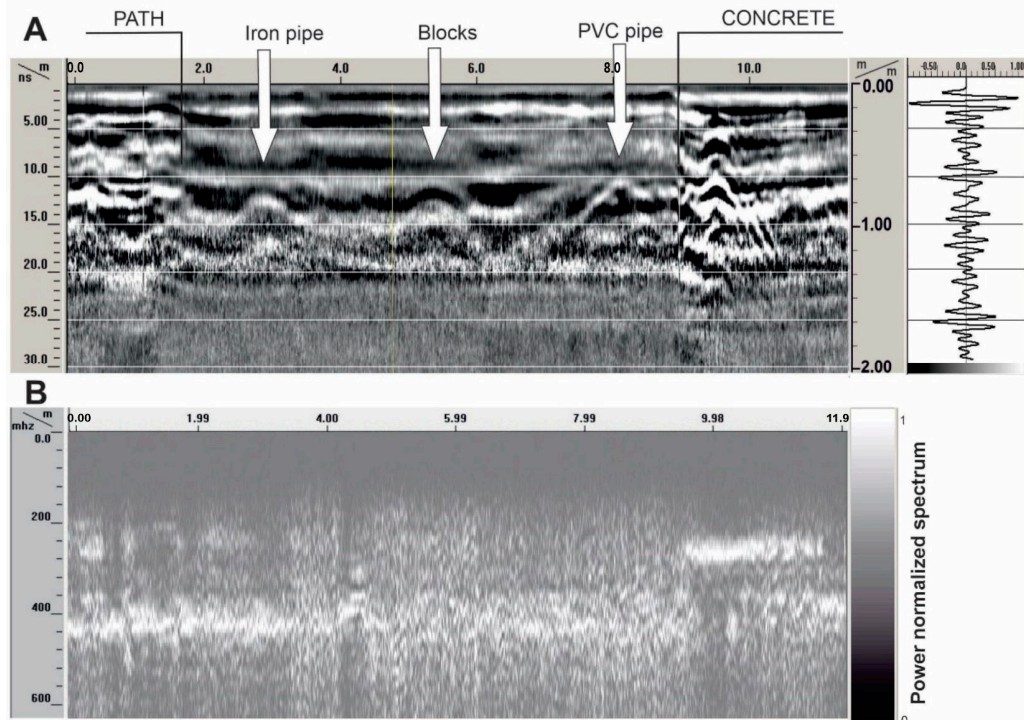

**Figure 8.** (**A**) Resulting B-scan after the processing flow of Table 2 was applied. X-axis represents traces, and Y-axis represents depth in meters and nanoseconds. The three buried structures are marked with arrows. The image's right side depicts the A-scan (trace) indicated by the light-yellow line on the B-scan (radargram). (**B**) Corresponding spectral normalized power. X-axis represents traces, and Y-axis represents the frequency in megahertz.

### 2.4. Analysis of Site 1 Using 3D-GPR to Enhance the Coherence Signal

Figure 9 shows the depth slices for site 1 up to the mean depth of 20 ns, i.e., 0.65 m depth, considering a dielectric constant of 20, obtained from the hyperbola analysis for migration and confirmed by the velocities used to reach the buried features using the plots described in Figure 6. The successive images show the evolution of the buried elements, verifying that the three linear structures still present a low detection. It was also verified that the PVC pipe and the eastern end of the row of bricks were first detected at a depth of 40 cm, possibly caused by irregularities in the base of the trenches ($\pm$10 cm). Figure 9A emphasizes the buried elements, and the other reflections observed in the slices are related to concrete slabs with a structural mesh that masks the medium, as they are highly reflective. It can be seen that the buried blocks have very low reflectivity, and the pipes show hardly any amplitude contrast, as the signal was attenuated in the shallower centimeters.

Furthermore, the normalized B-scan amplitudes underwent a multiplication by a factor of 10,000 when generating slices from a three-dimensional volume. The processing software converts decimal numbers to integers in order to facilitate their manipulation. Consequently, the range of amplitudes displayed in the slices, spanning from −10,000 to 10,000, does not possess a corresponding physical unit.

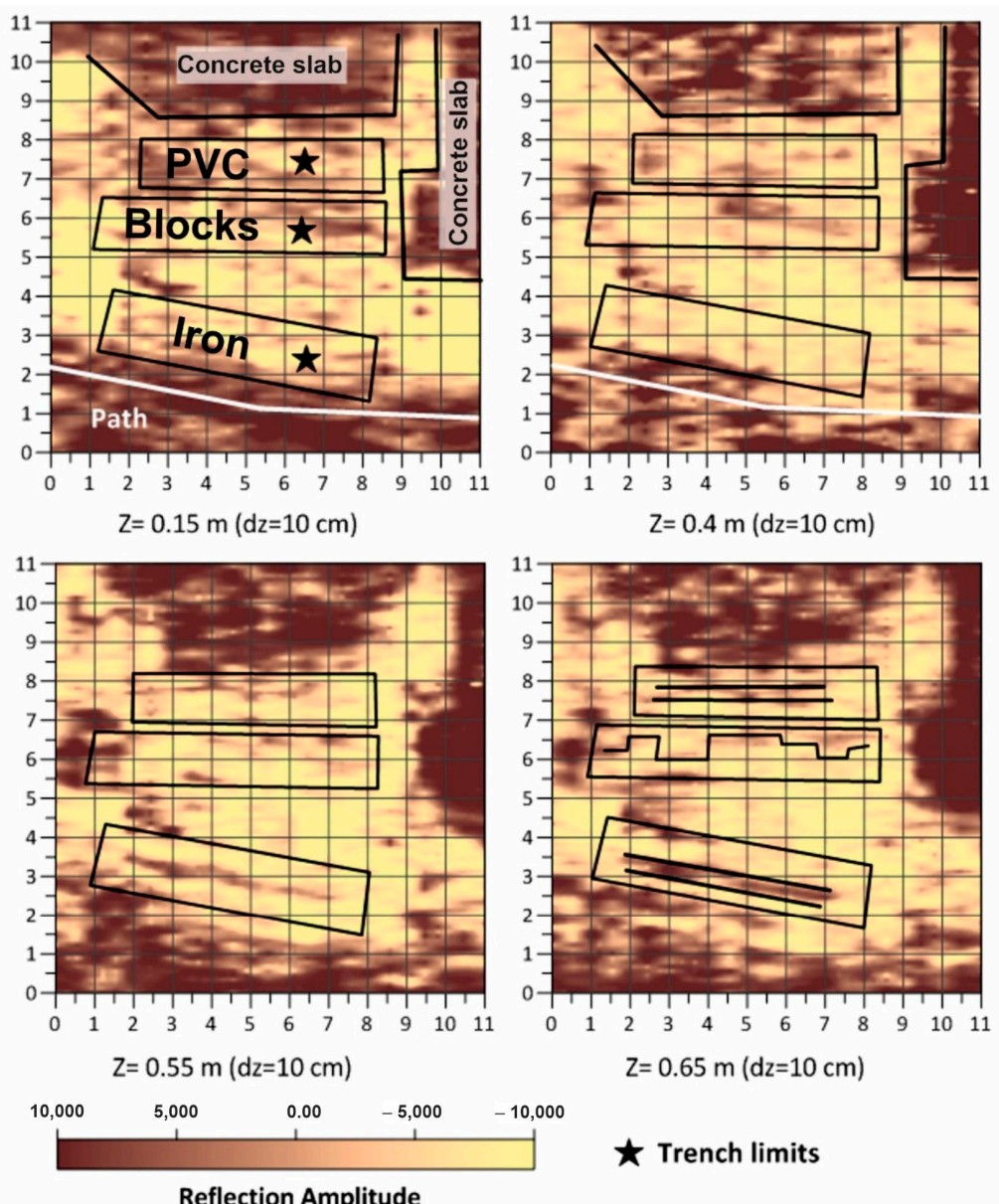

**Figure 9.** Slices at different depths to see the evolution of the buried elements in site 1. The top of the PVC pipe and some precast blocks were first detected at a depth of 0.4 m; the grid units are in meters. Rectangles highlight piperlines.

In order to obtain a greater continuity of these three elements, the cover surfaces method [39] was applied considering three slices between 0.4 m and 0.65 m. 3D GPR Cover surfaces is an alternative technique to construct 3D images that improve the visualization of GPR results. Broadly speaking, this technique consists of considering the reflections whose amplitudes exceed a pre-set threshold value, in our case A = 1000, for each slice, which refers to the maximum amplitude to be covered by the algorithm. The threshold value was selected after several test with values between the 50% and 85% of the maximum amplitude. Thus, when the amplitude of a mesh-point in the initial slice is below the selected threshold, it will not be retained in the 3D cover surface. On the other hand, if the amplitude is much higher than the threshold value, it will be retained in the cover surface and may appear as a bright spot or anomaly normalized to the threshold value established.

Thereby, the data mesh $(x_i, y_j)$ is traversed comparing the amplitudes $A_{ij}$ of each depth slice and considering the deepness of the first slice that meets the above condition.

Therefore, an image is obtained that represents the depth at which the roof of the different structures is obtained. Figure 10 shows the result obtained for site 1, where the structure with the lowest detection was the brick row. From our perspective, it can be observed that during the burial process of the blocks, the side holes of the bricks were spontaneously occupied by sediments from the host medium. This resulted in the attainment of nearly identical dielectric constants between the brick–clay and clay materials, primarily due to the presence of low-impedance contrasts. Figure 10 is a result of the cover surface analysis and demonstrates an improvement in the detection of the buried pipes and scattered reflections near the blocks.

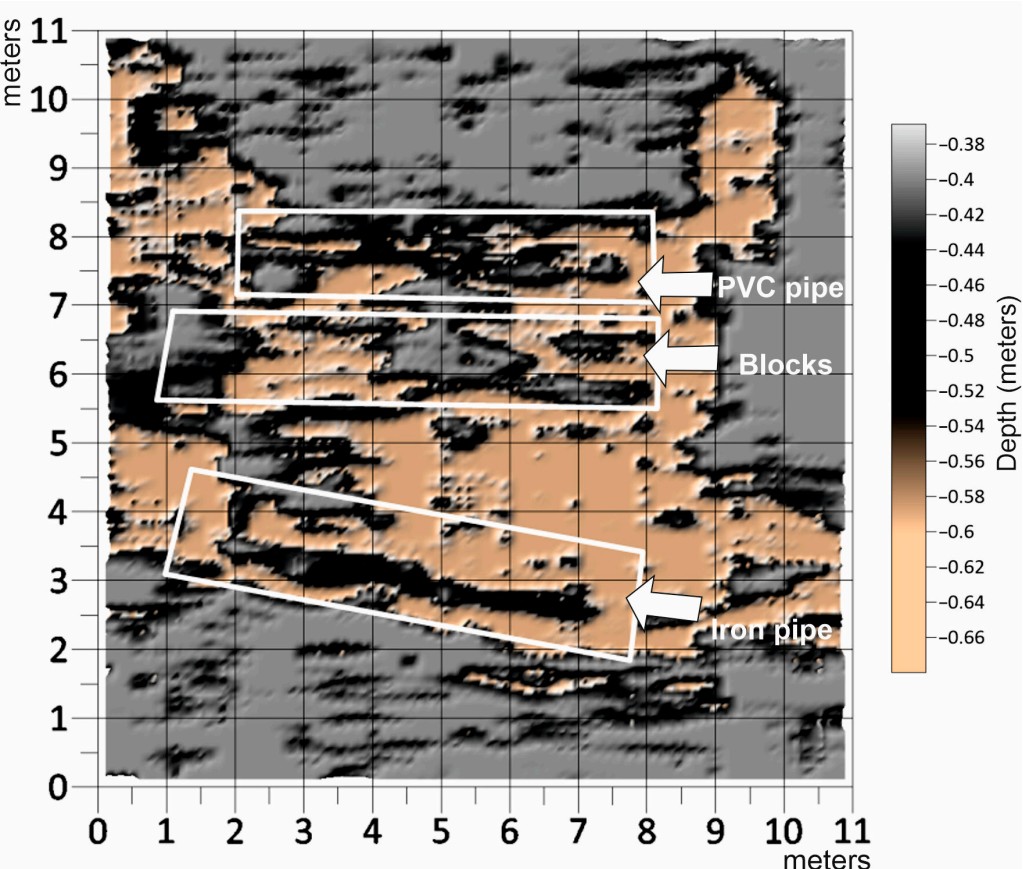

**Figure 10.** Result of applying the cover surface technique between 0.4 and 0.65 m depth at site 1. Using this method, it was possible to increase the continuity of the two pipes. White rectangles highlight piperlines.

### 3. Results in Site 2 and Discussion

The processing flow and the cover surface algorithms were applied in site 2 (Figures 1C and 3B). In this case, the relative dielectric constant was around 30, and the conductivity of the subsurface was 0.015 S/m (about 60 ohms*m), similar to site 1 (0.02 S/m, equal to 50 ohms*m), where the differences are based on the diverse geological environment (the lutitic subsoil was replaced, in this site, by marly facies), which in turn explains the skin-depth variations from one location to another. However, the GPR signals have similar features (Figure 11).

There is one aspect to comment on, which also occurs at site 1: when the ground surface is highly compacted, strong reflections can be produced in the interface of the subsequent layer. To reduce these multiples as much as possible, an additional algorithm was used, consisting of a horizontal filter that eliminates coherent signals longer than 3 m; the result is that the base of the compact layer is located around 15 ns on average, about 0.25 m deep with a dielectric constant of 15 (Figure 11C). These values give a skin-depth range from 1.5 to 2.1 m and attenuation coefficient from 3 to 7.5 dB/m (lower than

previous stage). The conductivity value of 0.015 S/m gives an approach of 4.5 dB/m for the attenuation coefficient.

**Figure 11.** Selected raw B-scan of site 2 (X = 3) with two buried structures (arrows). (**A**) The red rectangle highlights strong reflections produced by compacted path. (**B**) The spectral power of the same B-scan. (**C**) Resulting B-scan after the flow processing.

Two very clear hyperbolas at 3.5 m and 4.5 m from the origin of the GPR profile are shown in Figure 11A (profile position: X = 3 m). The first hyperbola was found at a depth of 0.3 m, and as seen in Figure 11, it can be correlated in all profiles. Its direction, which is parallel to the sidewalk, is detected in Figure 12. Given the strong dielectric contrast that it offers, it could correspond to a metallic pipe, which suggests a drinking water supply. The second of the hyperbolas (at 4.5 m from the origin in Figure 11) is at a similar depth (approximately 0.40 m). This hyperbola is correlated from x = 0 m to x = 6 m, and it could be a branch of the main metallic pipe towards the plots located in that section of sidewalk. It can be deduced from Figure 12. Therefore, it would be associated with the drinking water supply.

After additional processing was performed to remove the multiples associated with the compacted layer, two new hyperbolas were detected in the profile X = 3 (Figure 11C). The first, 10 m from the origin, is related to the trench made to install the central sewer line. According to the direct information taken in the outcrop (from the well that is accessed from one of the old manholes), the pipe should have been found at about 1.5–2 m depth. However, given its depth, it could not be detected in our study (only the excavation trench associated with it could be detected).

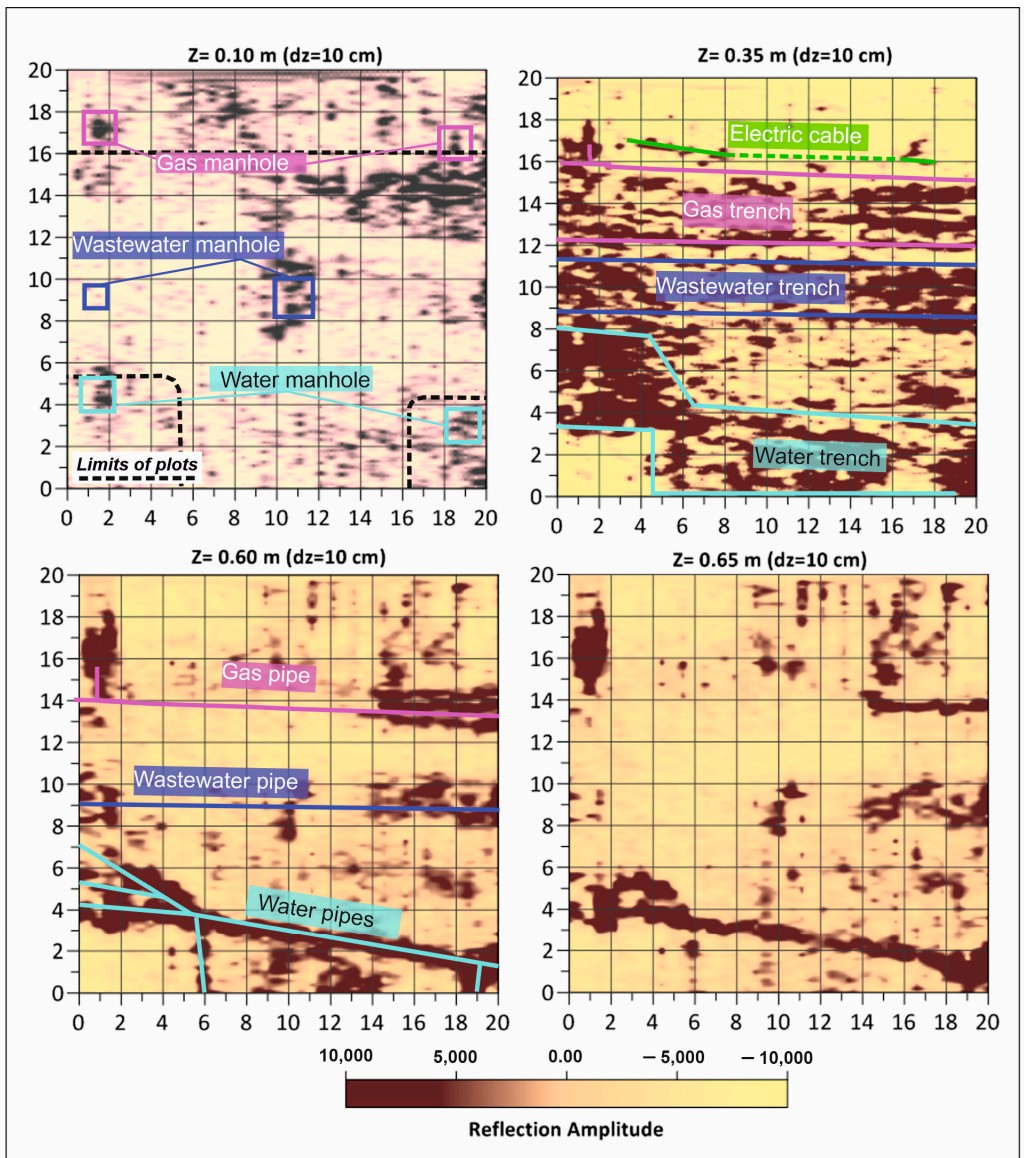

**Figure 12.** Selected depth slices at different depths to see the constructive evolution of the buried elements of site 2; the grid units are in meters.

Between 14–16 m from the origin, a new hyperbola appeared at a depth of 0.30 m, and it showed good continuity in the rest of the profiles and in the slices made (Figure 12). This hyperbola could also correspond to a pipe, although in this case, given the low dielectric contrast, it may be non-metallic (probably polyethene) and destined for another type of supply (probably gas). On the surface, approximately 18 m from the origin, the profile passes over a small trench in the direction of the "x"-axis, where the remains of corrugated wire loom tubing can be observed. This trench may be the result of the looting, from the nearby old manhole, of the wiring to provide electrical or telephone supply. For this reason, the wiring in the GPR profile was not detected with continuity (Figure 12).

The slices depicted in Figure 12 were utilized as input files to perform the cover surfaces shown in Figure 13A. The algorithm systematically assesses the amplitude of each $A_{ij}$ position within the data mesh grids corresponding to each slice. In this context, $A_{ij}$ indicates the matrix notation representing the data values within each slice, with $i$ and $j$ denoting the rows and columns that define the position of the data amplitude within the mesh. During this evaluation process, the algorithm compares all the $A_{ij}$ positions across the input slices and records the depth at which the highest amplitude is observed.

To ensure uniformity in the maximum amplitude values, a threshold of 1000 is established before executing the cover surfaces algorithm.

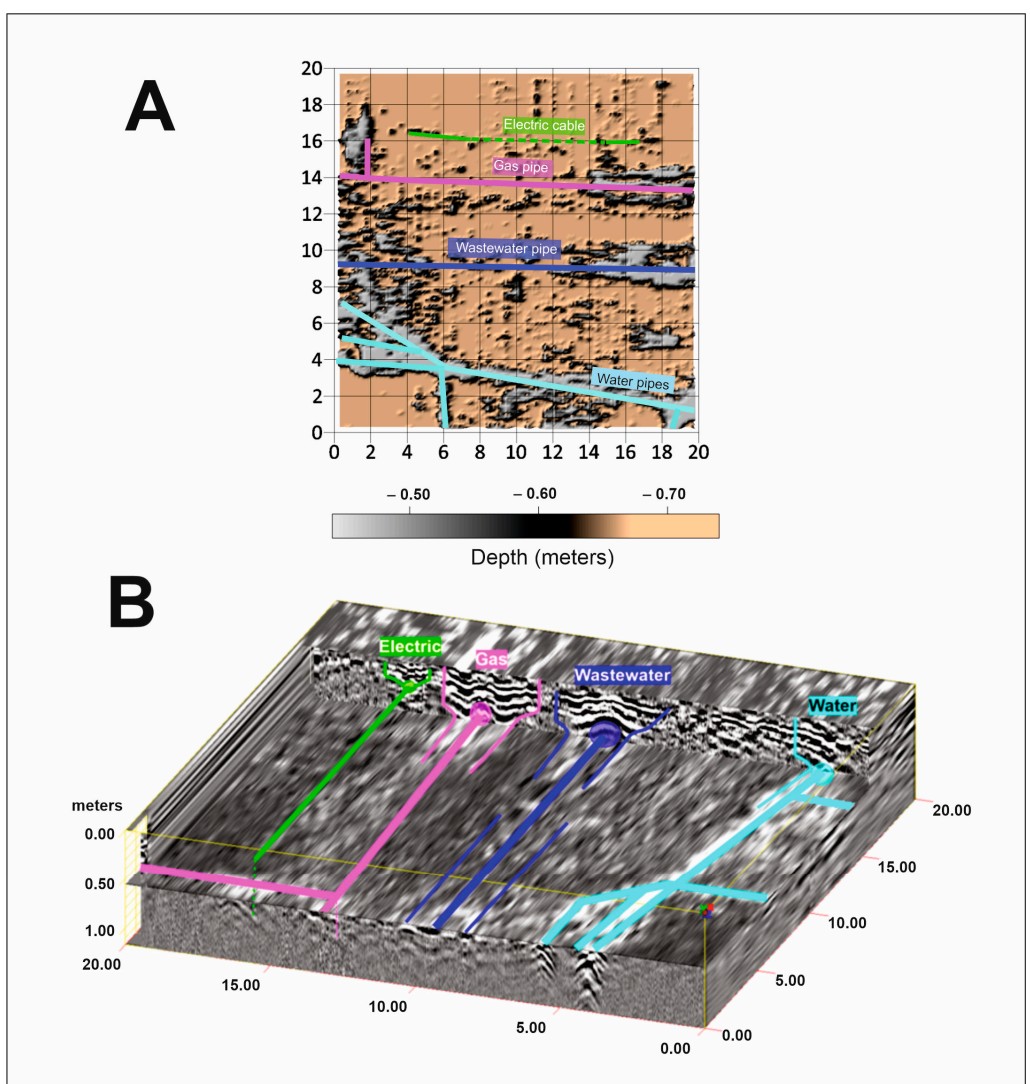

**Figure 13.** (**A**) Cover surface of site 2 up to the 0.65 m, coinciding with the mean of skin depth. (**B**) Three-dimensional view with the main pipelines placed at a depth of 0.5 m; the grid units are in meters.

Consequently, the amplitudes of the four slices for each position are meticulously traversed and compared. As a result, a consolidated representation emerges in the form of a new single slice, possessing the key features exhibited by the preceding four slices. This cover surface slice accentuates the amplitude of signals emanating from the pipes or other detectable elements such as walls while discarding low amplitudes to maintain continuity of the reflected materials. The cover surface representation encapsulates a comprehensive overview of the detected bodies within the substrate, adhering to preselected thresholds. This enables a more precise interpretation of the findings, particularly in regions where signal attenuation is evident, as observed during the investigation process. Figure 13B represents a 3D image of site 2. The X/Y plane is represented at a depth of 0.50 m, showing all the pipes buried in the subsoil. The X/Z plane is at a distance of 18 meters from the origin, and it represents the trenches of each element in depth, as it is able to deduce the lateral limit of each trench.

## 4. Conclusions

In conductive media, the detection of structures is poor. This factor must be accounted for when defining a GPR survey aimed at detecting buried utilities in civil engineering. In this sense, in clayey substrate, the results may not be as expected.

We carried out a controlled experiment (site 1) to establish an optimal processing flow in order to recover the GPR signal as much as possible without creating artifacts. In this first site, consisting of a subsoil composed of highly conductive clayey facies, different structures were buried at a known position and depth in order to be used for calibration purposes during the experiment.

This optimal processing flow was divided into two stages: 2D processing of the profiles and 3D processing. The key step of the 2D flow was the predictive deconvolution since with them, most of the multiples caused by other structures were eliminated. It was possible to emphasize the signals of interest by reducing the surrounding signals as much as possible. In order to obtain a greater continuity of these three elements (3D processing), the cover surfaces method was applied. This technique involves an analysis of reflections with amplitudes exceeding a predetermined threshold value across a sequence of consecutive depth slices. This approach yields enhanced visual outcomes, thereby facilitating the interpretation process in environments characterized by signal attenuation, such as clay formations in the site 1. Once this processing flow was established, a GPR exploration was carried out in an old, abandoned urbanization, and the same flow was applied. The use of 2D flow defined in site 1 allowed visualizing some structures that were not detected by applying preliminary processing flow. On the other hand, the cover surfaces algorithms were also applied in site 2, where they made it possible to highlight the continuity of some of the structures.

Although the GPR technique has great limitations for studies of conductive subsoils (0.02 S/m in site 1 and 0.015 S/m in site 2), the obtained results show the improvement in detecting and defining the elements buried in the subsoil due to optimized processing determined by knowing the type of structure, its dimensions, and the position and depth at which it is located.

**Author Contributions:** J.R. conceived the initial ideas of the study; B.M., R.M. and J.R. participated in the field survey; C.A.-P. and J.R. processed and analyzed the data; R.M., C.A.-P., B.M., J.R. and M.C.H. contributed to the data interpretation; R.M., B.M., J.R. and M.C.H. contributed to the preparation of the manuscript. All authors have read and agreed to the published version of the manuscript.

**Funding:** This work has been financed by the FEDER Andalucía R+D+i Project (reference 1380520), the Ministerio de Ciencia e Innovación Project (reference PID2021-123506OB-I00) and by the University of Jaen's own funds.

**Data Availability Statement:** The data that support the findings of this study are available from the corresponding author upon reasonable request.

**Acknowledgments:** The authors thank T. Teixido and J.A. Peña for the help provided during the course of the research and writing. In addition, the GuidelineGeo Company provided the information of the Mala 500 Mhz antenna, which was necessary to carry out the study.

**Conflicts of Interest:** The authors declare no conflict of interest.

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
