# Peer review of "Processing GPR Surveys in Civil Engineering to Locate Buried Structures in Highly Conductive Subsoils"

_remotesensing, doi:10.3390/rs15164019_

Round 1

Reviewer 1 Report (Previous Reviewer 1)

My comments on the initial version of the manuscript have been sufficiently addressed by the authors in this revised version. I have no further comments on the technical aspects. The manuscript may be considered for publication after a proofreading.

Author Response

We express our heartfelt thanks for all yours suggestions

Reviewer 2 Report (Previous Reviewer 2)

The reviewer thanks the authors for taking up the time to improve their manuscript. It seems that it reached a good level, therefore, the reviewer suggests its publication as is.

Author Response

We express our heartfelt thanks for all yours suggestions. 

Reviewer 3 Report (Previous Reviewer 3)

Review of amended “Processing GPR surveys in civil engineering to locate buried 3 structures in highly conductive subsoils”.

The authors have responded to the previous comments made and made some modifications to the paper.

There are still significant issues with the paper which need to be addressed.

The radar signal reflected from any buried target is dependent on a number of factors which are as follows.

1              Spreading loss which in a  free space environment is dependent in power terms on R-4 and voltage terms R-2.

2              Coupling loss from free space into the soil

3              Coupling loss from the soil into free space

4              Antenna gain. This must be qualified by the overall bandwidth of the antennas rather than at a single frequency which is inadequate as a full descriptor as is the radiation pattern at a single frequency. In addition, the bistatic antenna patterns need to be considered and calibrated not just the monostatic characteristic.

5              EM absorption loss in the soil which is best described in relation to the loss tangent of the soil and not the conductivity. It will be noted that  . The dielectric constant of a material has two components real and imaginary. In essence the loss in the material can be due to finite material conductivity as well as dielectric damping. In general, it is only possible to distinguish the contribution to the measured imaginary part of the permittivity by considering the experimentally derived imaginary permittivity at two different frequencies. Alternatively, the material loss can be determined by taking measurements using a calibrated target buried at known depth or using antennas inserted into the soil to measure the path loss.

6              Target radar cross section which depends on shape and impedance difference between the target and host material.

Measurement of buried targets by a GPR effectively sums all these losses together, so the signal received before processing cannot be assumed to be a measurement of soil losses alone. Unless all these parameters are quantified it is not possible to determine anything absolute. All that can be postulated is that there is a signal return but what exactly is indeterminate. The authors must address this issue as their results are simply relative rather than absolute and hence nothing useful can be concluded from the paper that is not already known.

The authors state that the antenna has  a footprint of 4 cm at 6 m depth.”   This value does not make any sense. Fig 4 A is not relevant to the paper and should be removed. Fig 4 B and C should be measured bistatic patterns, as this is what was used to carry out the study, not an idealised value which is presumably a free space pattern rather than the pattern in the soil which can be radically different.

The authors state that the GPR processing has been carried out using estimates of attenuation losses for clay and marl of 5-9dBm-1 at is presumed a frequency of 500 MHz. This is radically lower than values given in the literature of between 15-50 dBm-1 and leads to concerns over the methods for estimating losses described by the authors. In essence the results appear to be outliers compared with the known literature and hence are questionable.

The plots shown in Fig 7b and 8b need further explanation. If the plots are correlated with the B Scan time plots, then why is spectral power highest at positions lower than the targets when it is known that spectral power will reduce as a function of depth.?

Plots 9 and 10 show the difficulty of concluding much without a priori knowledge  of the buried targets, as a significant proportion of the image is contaminated with other unexplained reflections.

It is recommended that all plots are shown as grey scale and the colour plots are changed as the colour confuses the assessment of the plots.

The authors have not really addressed the key issues and the paper lacks scientific rigour, exhibits weak experimental practice and hence it is unsuitable for publication.

The authors may wish to read more on the physics of soil parameters and radar from the following.

Hoekstra, P., Delaney, A.,

Dielectric properties of soils at UHF and microwave frequencies

J. Geophys. Res., Vol 79, 1974, pp 1699-1708

Campbell, M.J., Ulrichs, J.,

Electrical properties of rocks and their significance for lunar radar observations

J. Geophys. Res., Vol 74, 1969, pp 5867-5881

Von Hippel, A.R., (Ed)

Dielectric materials and applications

MIT Press, Cambridge, Mass., U.S.A, 1954, p 314

Hipp, J.E.,

Soil electromagnetic parameters as functions of frequency, soil density and soil moisture

Proc. IEEE, Vol 62, No 1, 1974, pp 98-103

Cole, K.S., Cole, R.H.,

Dispersion and absorption in dielectrics - I Alternative current characteristics

J. Chem. Phys., Vol 9, 1941, pp 341-351

De Loor, G.P.,

The dielectric properties of wet materials

IEEE Trans., Vol GE-21, 1983, pp 364-369

De Loor De Loor, G.P.,The dielectric properties of wet materials IEEE Trans., Vol GE-21, 1983, pp 364-369

Hoekstra, P., Delaney, A., Dielectric properties of soils at UHF and microwave frequencies. Geophys. Res., Vol 79, 1974, pp 1699-1708

Hipp, J.E., Soil electromagnetic parameters as functions of frequency, soil density and soil moisture Proc. IEEE, Vol 62, No 1, 1974, pp 98-103

Wang, J.R., Schmugge, T.J., An empirical model for the complex dielectric permittivity of soil as a function of water content IEEE Trans., Vol GE-18, 1980, pp 288-295

Hallikainen, M.T., Ulaby, F.T., Dobson, M.C., Elrayes, M.A., Wu, L.K., Microwave dielectric behaviour of wet soil Parts I and II, IEEE Trans., Vol GE-23, 1985, No 1, pp 25-34

Wobschall, D.,A theory of the complex dielectric permittivity of soil containing water: The semidisperse model IEEE Trans., Vol GE-15, 1977, pp 49-58

Neil R. Peplinski, Fawwaz T. Ulaby, and Myron C. Dobson

Dielectric Properties of Soils in the 0.3-1.3-GHz Range

IEEE TRANSACTIONS ON GEOSCIENCE AND REMOTE SENSING, VOL. 33, NO. 3, MAY 1995 803

Miller, T.W., B. Borchers, J.M.H. Hendrickx, Sung-Ho Hong, L.W. Dekker, and C.J. Ritsema, 2002. Effects of soil physical properties on GPR for landmine detection. Proceedings of the Fifth International Symposium on Technology and the Mine Problem, 10 pp.

-

Minor

Round 2

Reviewer 3 Report (Previous Reviewer 3)

The authors have responded to the previous comments but have not really addressed the issues. Experienced users of GPR are far more familiar with the concepts of attenuation losses rather than skin depth yet the authors persist in using a descriptor that is not in frequent use and this is confusing. The authors state It is important to notice that attenuation coefficient and losses are related concepts in ground-penetrating radar (GPR) systems, but they represent different aspects of signal propagation and signal strength reduction in the subsurface medium.  This is just not correct.  The key point is that the paper does not include properly measured data on the soil attenuation and the equation given in the paper is not a full description. The use of multiple terms in the paper ie skin depth, attenuation coefficient, attenuation loss is confusing and the readers would find the use of one term clearer. To aid the authors I have attached the standard treatment of material losses but they still need to measure these properly for the paper to have any merit

Author Response

This manuscript is a resubmission of an earlier submission. The following is a list of the peer review reports and author responses from that submission.

Round 1

Reviewer 1 Report

I noticed that this is a revised and resubmitted manuscript. The author has done a good job with the revisions. In addition, I still have a few remaining further questions.

1.      In Introduction: This manuscript should further add some articles and will be of interest to many, such as:

(1)  Performance Evaluation of Full-scale Accelerated Pavement using NDT and Laboratory Tests: A case study in Jiangsu, China: https://doi.org/10.1016/j.cscm.2023.e02083.

(2)  Automatic recognition of pavement cracks from combined GPR B-scan and C-scan images using multiscale feature fusion deep neural networks. https://doi.org/10.1016/j.autcon.2022.104698.

2.      There seems to be a problem with the formatting of Equation (2), Table 2 and Figure 5. Please check.

Minor editing of English language required

Author Response

Dear Reviewer 1

We have prepared a new revised version of the manuscript. The modifications are highlighted (texte110723.docx). Please find below our answers to the comments and requests.

  1. In the Introduction section, we have extended the review of the literature on the use of GPR in Civil engineering: (1) Performance Evaluation of Full-scale Accelerated Pavement using NDT and Laboratory Tests: A case study in Jiangsu, China: https://doi.org/10.1016/j.cscm.2023.e02083. (2) Automatic recognition of pavement cracks from combined GPR B-scan and C-scan images using multiscale feature fusion deep neural networks. https://doi.org/10.1016/j.autcon.2022.104698.
  2. Format problems have been checked and fixed

Reviewer 2 Report

Processing GPR surveys in civil engineering to locate buried 2 structures in highly conductive subsoils

Revision of the revised paper

Although the authors attest that changes have been made in the abstract, it seems that it maintained the same writing. Regarding the last statement “The results obtained show the validity of the proposed treatment.”, the reviewer asks again what are these results? What was the overall processing applied to the field/lab data?

The part regarding skin depth is still not very clear.

Scales in figure 5/6 do not appear in this version, apparently. The reviewer suggested a graphical scale, which does not appear.

The justification for filling of brick’s voids is plausible.

The amplitude of 1000 is still not justified for the covers method. The point here is that the authors do not justify the number. It seems random. And what happen in cases the amplitudes are below this level, or much higher?

The conclusions are almost the same, so the reviewer maintains the same opinion. The results are not validated. It is a fact that high conductive media are very difficult to assess by GPR. The method used is nothing new, just to make averages and thresholds to better visualize the field data. Additionally, most of the sector 2 experiments are based on external clues and knowledge about the way infrastructure develop to interpret the resulted radargrams. Which is what most GPR operators use for interpretation.

Although a lot of new information and improvement was carried out, the validity of the method is not proven, and this should be removed from the article conclusions. Also, the authors should look at this as a technical experiment, maybe not as a scientific paper.

Therefore, I maintain the Major Revision.

Moderate english revision necessary.

Author Response

Dear Reviewer 2

We have prepared a new revised version of the manuscript. The modifications are highlighted (texte110723.docx). Please find below our answers to the comments and requests.

Reviewer's comment and request:

“Although the authors attest that changes have been made in the abstract, it seems Format problems have been checked and fixedthat it maintained the same writing. Regarding the last statement “The results obtained show the validity of the proposed treatment.”, the reviewer asks again what are these results? What was the overall processing applied to the field/lab data?

The conclusions are almost the same, so the reviewer maintains the same opinion. The results are not validated. It is a fact that high conductive media are very difficult to assess by GPR. The method used is nothing new, just to make averages and thresholds to better visualize the field data. Additionally, most of the sector 2 experiments are based on external clues and knowledge about the way infrastructure develop to interpret the resulted radargrams. Which is what most GPR operators use for interpretation.”

Authors' response:

Thank you very much for your support. I agree to remove the phrase "the validity of the method", because it would need more arguments. In response to the comments, the conclusions of the work added in the summary and in the conclusions have been modified. Therefore, the abstract and conclusions only include a discussion of the results obtained.

Reviewer's comment and request:

“The part regarding skin depth is still not very clear”

Authors' response:

This aspect of the skin depth was clarified in the main manuscript for the final version.

Reviewer's comment and request:

“Scales in figure 5/6 do not appear in this version, apparently. The reviewer suggested a graphical scale, which does not appear.”

Authors' response:

The scales were added to all figures.

Reviewer's comment and request:

The justification for filling of brick’s voids is plausible.”

Authors' response:

When the blocks were buried, it is presumed that their voids were filled with host material, since they were not clearly identified in comparison with the pipes.

Reviewer's comment and request:

“The amplitude of 1000 is still not justified for the covers method. The point here is that the authors do not justify the number. It seems random. And what happen in cases the amplitudes are below this level, or much higher?”

Authors' response:

This value was selected after having performed several tests with values between 50% and 85% of the maximum amplitude, being the value 1000 the one that gave the best results in indicating the continuity of the pipes horizontally with the least possible noise generated by the host medium. This argument has been added to the manuscript.

Reviewer's comment and request:

“Although a lot of new information and improvement was carried out, the validity of the method is not proven, and this should be removed from the article conclusions. Also, the authors should look at this as a technical experiment, maybe not as a scientific paper.”

Authors' response:

The affirmation of the validity of the method has been eliminated, although the improvement of the results obtained by an optimized flow of data treatment is maintained. The manuscript is as a technical note experiment, not as a scientific article.

Reviewer 3 Report

The authors do not appear to have attended to the previous comments which are substantially still relevant.

The paper needs to include a proper physical measurement of the soil attenuation and soil relative dielectric constant over the range of frequencies concerned, that is from 50 MHz to 2 GHz. The method of estimating the skin depth described in the paper is not a good approach in that the losses of the radar measurement are not properly quantified; hence the paper is not useful and is basically a subjective interpretation of radar images many of which fail to show amplitude scaling.

 The soil attenuation loss should be determined using a proper method of which many are well described in the literature and enough measurements should be taken to provide statistical confidence in the value measured. The lack of this information is a major weakness of the paper.

The authors need to include in their assessment; values of GPR coupling losses, spreading loss, attenuation loss in dB per metre, target reflection loss, target radar cross section [as a function of polarisation] all as a function of frequency. There is no information on antenna radiation patterns which essentially contribute to gain and hence affect the estimates of path losses.

There is no information on the GPR system loop gain which would serve as a means of cross-checking path losses.

Proper absolute scales missing on figs 5,6,7,8,9,,10, 11

Specifically what is meant by reflection amplitude 10000 to -10000? What units - voltage? power?- normally reflection is measured in either linear [volts] or log scales [dB]

Minor typos

Author Response

Dear Reviewer 3

We have prepared a new revised version of the manuscript. The modifications are highlighted (texte110723.docx). Please find below our answers to the comments and requests.

Reviewer's comment and request:

“The paper needs to include a proper physical measurement of the soil attenuation and soil relative dielectric constant over the range of frequencies concerned, that is from 50 MHz to 2 GHz. The method of estimating the skin depth described in the paper is not a good approach in that the losses of the radar measurement are not properly quantified; hence the paper is not useful and is basically a subjective interpretation of radar images many of which fail to show amplitude scaling.”

Authors' response:

Thank you for your suggestion. Unfortunately, in this case, we are unable to conduct the requested experiment due to limitations with our single-channel antenna. Our antenna's radiation lobe is focused specifically at 500 MHz, and we are unable to extend beyond the capabilities of the GPR antenna. Furthermore, there are side radiations at 200 MHz and 400 MHz, but their power levels are significantly low, which prevents us from fully carrying out the experiment you have proposed. Typically, experiments of this nature are conducted using different antennas or GPR step-frequency arrays that can sweep frequencies ranging from 40 MHz to 3 GHz. Regrettably, the scope of our work does not encompass the specific requirements outlined in this request.

 Reviewer's comment and request:

“The soil attenuation loss should be determined using a proper method of which many are well described in the literature and enough measurements should be taken to provide statistical confidence in the value measured. The lack of this information is a major weakness of the paper.”

Authors' response:

Thank you for your valuable insights; our first approximation to determine the losses is the attenuation coefficient, which is inversely proportional to the skin Depth. This approach gives us the required information in Nepers/m. Regardless, we have calculated signal loss in dB/m from our first approach.

Reviewer's comment and request:

“The authors need to include in their assessment; values of GPR coupling losses, spreading loss, attenuation loss in dB per metre, target reflection loss, target radar cross section [as a function of polarisation] all as a function of frequency. There is no information on antenna radiation patterns which essentially contribute to gain and hence affect the estimates of path losses.“

Authors' response:

Regrettably, the execution of these procedures falls outside the purview of the undertaken research experiment and is commonly accomplished via laboratory simulations utilizing synthetic data, which can subsequently be juxtaposed with field observations. The aforementioned calculations can be further explored in the subsequent phase of our research. The present discussion pertains to the radiation patterns of a dipole single-channel antenna that has been modified with a bow-tie design. In this particular setup, the receiver and transmitter of the antenna are positioned at a distance of 40 cm from each other within the same device. This arrangement ensures that the lambda/2 condition is satisfied for frequencies of 500 MHz (corresponding to a wavelength of 0.3 m). Hence, the manufacturer calibrates the antenna to concentrate its radiation power at 500 MHz within the main frontal lobe, achieving the lambda/4 condition, achieving a radial resolution of 0.04 m up to 6 m depth with comparatively reduced energy in the sidelobes. The values in question are established by the manufacturers and align with the principles of Ground Penetrating Radar (GPR) theory, as stated in the antenna's device manual. Therefore, we have a high level of confidence in the default electronic configuration of the device. The relevant information is contained within the manuscript. We appreciate your advice.

Reviewer's comment and request:

“There is no information on the GPR system loop gain which would serve as a means of cross-checking path losses.”

Authors' response:

Unfortunately, this information, although having some relevance to the antenna design and electronics, is beyond our scope. These could be considered in a simulation with the same conditions as the experiment, and then compared with the field data, but this would be a second part of further research.

Reviewer's comment and request:

“Proper absolute scales missing on figs 5,6,7,8, 9,10, 11”

Authors' response:

The proper scales were added to the manuscript figures, thanks to the advice.

Reviewer's comment and request:

“Specifically what is meant by reflection amplitude 10000 to -10000? What units - voltage? power?- normally reflection is measured in either linear [volts] or log scales [dB]”

Authors' response:

The range of reflection amplitudes is only a factor of 10000 applied by the processing software to the normalized amplitudes between -1 and 1 in the B-scans to construct the 3D volumes with integer amplitude values, making more easily to show the amplitude values in the displayed slices. The normalized values between -1 and 1 are normalizations made for better graphical representation, and therefore relate to the maximum and minimum amplitudes. These values are lacking in physical dimension, no power nor voltage units.  This information was added to the manuscript. Thanks for your advice.

Round 2

Reviewer 2 Report

The authors answered all my questions.

The reviewer suggests including a flow chart with the processing steps in a condense way.

Several small errors were detected. A careful revision should be undertaken.

Author Response

Dear Reviewer 2,

First at all, we thank you very much for your comments.

Following your suggestions, we have included a flow chart with the processing steps (new Figure 5 in the revised manuscript).

In addition, as you recommended us, we have carefully checked the entire text for English usage, with especial attention to the new added paragraphs.

Reviewer 3 Report

The authors explain that they are unable to provide the information required to properly quantify the work described in their submisssion.  Consequently the paper does not add in any significant way to an understanding of the capability of GPR as a means of detecting targets in lossy soils. The GPR is effectively uncalibrated and the soil attenuation as a function of frequency is guesstimated rather than measured. These are fundamental flaws in the study and the paper. The soil parameters could have been measured using a Vector Network Analyser which is standard laboratory equipment, calibrated wideband antennas and canonical targets buried at different depths in the soil. The GPR could have been calibrated using a canonical target [sphere or plate] in free space to provide a amplitude/ range measure of its performance. Low frequency EM conductivity is not a reliable indicator of soil loss in the frequency range from 50MHz to 5GHz and as attenuation in soils in directly proportional to frequency then Fig 4 is essentially meaniningless without reference to frequency.

The authors have not carried out this work in a rigourous way and the submission adds little to the overall body of knowledge of EM soil investigation.

Author Response

Dear Reviewer 3,

First at all, we thank the detailed analysis that you devoted to our paper and we really appreciate the interest of your suggestions about different approaches to improve our study. 

We would like to explain how we have dealt with the specific aspects that you mention in your comments.

Regarding the considerations on the antenna calibration, the antenna was calibrated by the manufacturer MALA Geoscience, and that should assure its correct operation. We understand your concerns about this fact, perhaps motivated by the absence of information about the radiation patterns of the antenna in our previous versions. So, we query to the manufacturer about this issue. The technical team of MALA Geoscience gave us several controlled tests performed to this device to fulfill the requirements of the American Normative FCC.15.209, FCC.15.509 and ANSI C63.4. Based on these tests, the antenna has a radiation pattern around 500 MHz, with 361 MHz 3dB bandwidth and 40 dB on its central lobe, with a pulse of 10.4 ns that reaches 6 m depth and a footprint of 4 cm in ideal conditions. Hence, we include in the reviewed manuscript information about the antenna test, and a graph with frequencies and emissions from one of the several tests performed on this equipment, which should not exceed 45 dB. With this info, and complemented with the spectrum frequency of the antenna spanning from 200 MHz to 700 MHz, we constructed an idealized radiation pattern, shown in Figure 4 of the revised manuscript. This information is included in a new subsection 2.2. Equipment, added to the section 2. Materials and Methods.  

Referred to the calculation of the parameters, the first thing what we do is to obtain the host media velocity, and from this value we could get the relative dielectric permittivity. Then, we confirm and adjust this calculation with the velocity’s hyperbolic analysis, that gave us similar velocities and, in consequence, similar relative dielectric permittivity values. These values are corrected due to the known position of the buried pipes of the first site, that serve as calibration factor in our experiment calculations. Then, we obtain the other values using the equations described in the manuscript, such as the skin depth (δ) and attenuation coefficient (α). A new paragraph was added to include this explanation in section 2.3 GPR data analysis.

On the other hand, you mentioned your concern about former Figure 4 not having reference to frequency. According to Cassidy (2009), the skin depth and the attenuation coefficient in the range of 50 MHz to 5 GHz do not depend on the frequencies but rather on the media conditions (permittivity, conductivity, and permeability). Thus, the plots shown in the former Figure 4 (now Figure 6) do not depend on the frequency range, although frequency is a fundamental factor in the depth resolutions achieved by an electromagnetic emission.  

So, when we refer to a “low frequency EM conductivity” in our manuscript we are considering it in the range of GPR conditions (frequencies greater than 10 MHz), and, therefore, the plots shown can be made without the need to refer to the frequency. Maybe this was not clearly presented in our paper. Thus, we have specifically indicated in the revised manuscript that these values have been calculated with the second condition of the equation 5 mentioned in the text, and that the frequency of the central lobe is centered around 500 MHz, to avoid confusion in this respect.

We hope that our explanations fulfil your queries.
